# The Mediterranean Diet in Pregnancy: Implications for Maternal Brain Morphometry in a Secondary Analysis of the IMPACT BCN Randomized Clinical Trial

**DOI:** 10.3390/nu16111604

**Published:** 2024-05-24

**Authors:** Ayako Nakaki, Yvan Gomez, Sara Castro-Barquero, Allegra Conti, Kilian Vellvé, Irene Casas, Mariona Genero, Lina Youssef, Laura Segalés, Leticia Benitez, Rosa Casas, Eduard Vieta, Nuria Bargallo, Nicola Toschi, Ramon Estruch, Fàtima Crispi, Eduard Gratacos, Francesca Crovetto

**Affiliations:** 1BCNatal|Fetal Medicine Research Center (Hospital Clínic and Hospital Sant Joan de Déu), University of Barcelona, 08028 Barcelona, Spain; 2Institut d’Investigacions Biomédiques August Pi i Sunyer (IDIBAPS), 08036 Barcelona, Spain; 3Department of Internal Medicine Hospital Clinic, IDIBAPS, University of Barcelona, 08036 Barcelona, Spain; 4Centro de Investigación Biomédica en Red de Fisiopatología de la Obesidad y Nutrición (CIBEROBN), 28029 Madrid, Spain; 5Institut de Recerca en Nutrició i Seguretat Alimentaria (INSA-UB), University of Barcelona, 08028 Barcelona, Spain; 6Department of Biomedicine and Prevention, University of Rome “Tor Vergata”, 00133 Rome, Italy; 7Institut de Recerca Sant Joan de Déu, 08950 Esplugues de Llobregat, Spain; 8Josep Carreras Leukaemia Research Institute, Hospital Clinic/University of Barcelona Campus, 08036 Barcelona, Spain; 9Department of Psychiatry and Psychology, Hospital Clinic, Neuroscience Institute, IDIBAPS, University of Barcelona, 08036 Barcelona, Spain; evieta@clinic.cat; 10Centro de Investigación Biomédica en Red de Salud Mental (CIBERSAM), Instituto de Salud Carlos III, 28029 Madrid, Spain; 11Radiology Department, Center of Image Diagnostic, Hospital Clínic, Facultad de Medicina, Universidad de Barcelona, 08036 Barcelona, Spain; 12Athinoula A. Martinos Center for Biomedical Imaging, Harvard Medical School, Boston, MA 02115, USA; 13Institut d’Investigacions Biomèdiques August Pi i Sunyer (IDIBAPS), Barcelona, Spain and Centre for Biomedical Research on Rare Diseases (CIBERER), 08036 Barcelona, Spain; 14Primary Care Interventions to Prevent Maternal and Child Chronic Diseases of Perinatal and Developmental Origin, RD21/0012/0003, Instituto de Salud Carlos III, 28029 Madrid, Spain

**Keywords:** Mediterranean diet, pregnancy, magnetic resonance, cortical area, randomized clinical trial

## Abstract

Introduction: A Mediterranean diet has positive effects on the brain in mid-older adults; however, there is scarce information on pregnant individuals. We aimed to evaluate the effect of a structured Mediterranean diet intervention on the cortical structure of the maternal brain during pregnancy. Methods: This study was a secondary analysis of the IMPACT BCN, a randomized clinical trial with 1221 high-risk pregnant women randomly allocated into three groups at 19–23 weeks of gestation: Mediterranean diet intervention, a mindfulness-based stress reduction program, or usual care. Maternal brain magnetic resonance imaging was performed during the third trimester of pregnancy in a random subgroup of participants. For this study, data from the Mediterranean diet and usual groups were analyzed. Maternal dietary intake, adherence to the Mediterranean diet and metabolite biomarkers were evaluated using a food frequency questionnaire, a 17-item dietary screener and plasma/urine samples, respectively. Results: The cluster-wise analysis showed that the Mediterranean diet group participants (*n* = 34) had significantly larger surface areas in the right precuneus (90%CI: <0.0001–0.0004, *p* < 0.001) and left superior parietal (90%CI: 0.026–0.033, *p* = 0.03) lobules compared to the usual care group participants (*n* = 37). A larger right precuneus area was associated with high improvements in adherence to the Mediterranean diet, a high intake of walnuts and high concentrations of urinary hydroxytyrosol. A larger left superior parietal area was associated with a high intake of walnuts and high concentrations of urinary hydroxytyrosol. Conclusions: The promotion of a Mediterranean diet during pregnancy has a significant effect on maternal brain structure.

## 1. Introduction

During the past decade, research has gathered a lot of positive evidence on the Mediterranean diet’s effects on human health. A Mediterranean diet provides nutrients rich in bioactive compounds such as polyunsaturated fatty acids (PUFAs) and (poly)phenols, whose potential effects include free radical scavenging and anti-inflammatory properties. Estruch and colleagues conducted PREvención con DIeta MEDiterránea (PREDIMED), a randomized trial based on a Mediterranean diet supplemented with extra virgin olive oil and nuts in a population at high risk of cardiovascular disease [1] and demonstrated a decrease in major cardiovascular incidence [1], as well as an improvement in cognitive function in both arms on a Mediterranean diet compared to a low-fat diet [2]. In line with this effect of the Mediterranean diet on the adult brain, several studies with middle-aged or aged adults have demonstrated that high adherence to a Mediterranean diet has associations with an increased volume or thickness of the brain cortex in cognitively normal populations [3,4,5]. While some have described a larger total gray matter volume [4,6], there is a wide variety of regional areas reported in the literature, particularly the frontal lobe, parietal lobe, occipital lobe, posterior cingulate gyri and entorhinal cortex [3,6,7]. However, there is scarce information about the effects of a Mediterranean diet on the cortical morphometry in pregnant women.

Recent evidence has shown that during pregnancy, the maternal brain undergoes significant long-term neuroanatomical modifications [8], probably due to steroid hormone changes during this period [9,10]. In addition, there is a high demand for PUFA intake during pregnancy. The human brain is composed of 35–40% PUFAs, including 10% arachidonic acid and docosahexaenoic acid (DHA) [11], which are integral structural components of neurological systems. Their actions play a role in brain function by positively influencing neurogenesis [12] and neurotransmission [13] and promoting neurite growth and synaptic transmission [14]. The demand for PUFAs increases progressively throughout pregnancy, especially in the third trimester, to reach a need of 200 mg per day of DHA intake due to the accelerated neurological development of the fetus and the accumulation of unsaturated fatty acids in the nervous tissue [15]. Several data suggest that PUFA depletion or an imbalance between the n-6 and n-3 PUFA ratio in early pregnancy is associated with a range of neurological and psychiatric disorders in mothers [16] and the impaired neurodevelopment of fetuses [17]. The Mediterranean diet has the advantage of providing the necessary PUFA requirements through a diet mainly based on fish consumption and nuts. Thus, the Mediterranean diet can be considered an ideal diet during pregnancy. Indeed, some studies have evaluated the impact of the Mediterranean diet on adverse perinatal outcomes or offspring development [18,19,20,21], but none have assessed the maternal brain.

Recently, the IMPACT BCN (Improving Mothers for a better PrenAtal Care Trial BarCeloNa) randomized clinical trial showed that structured lifestyle interventions during pregnancy based on a Mediterranean diet or a mindfulness-based stress reduction program reduced the incidence of newborns born small for gestational age (SGA) and other adverse pregnancy outcomes [22], as well as supporting a different fetal brain structure [23] and the better neurodevelopment of infants at 2 years of age [24]. Moreover, the Mediterranean diet intervention was associated with a significant reduction in maternal anxiety and stress for the pregnant women [25]. We wondered whether such intervention could also influence maternal brain morphometry. As part of the IMPACT BCN trial, this study aimed to demonstrate whether a Mediterranean diet and its nutritional components affect the cortical structure in pregnant women.

## 2. Materials and Methods

### 2.1. Study Population and Design

The present study is a secondary analysis of a large randomized clinical trial, IMPACT BCN [22], conducted at a large referral center for maternal–fetal and neonatal medicine in Barcelona, Spain. The primary endpoint of the trial was the prevalence of SGA newborns, defined as having a birthweight below the 10th centile. The enrollment of the main study took place from February 2017 to October 2019 for 1221 individuals, with follow-up still ongoing (with the final follow-up for deliveries on 1 March 2020). The study population was women recruited at mid gestation (19–23.6 weeks) at high risk of having an SGA newborn, according to the criteria of the Royal College of Obstetrics and Gynaecologists [26]. All the individuals who agreed to participate provided written informed consent on the day of recruitment. The protocol was approved by the institutional review board (HCB-2016-0830, HCB-2020-0267), and the trial was registered on ClinicalTrials.gov (Identifier: NCT03166332).

Participants who agreed to take part to the trial were randomly allocated at a 1:1:1 ratio into three groups: a Mediterranean diet intervention group, a stress reduction program based on mindfulness techniques, or the usual care. The inclusion and exclusion criteria are reported elsewhere [22,27]. For this specific study, at their visit at 29–34 weeks of gestation, a sub-sample of 350 randomly selected participants was recruited for an MR assessment, which was planned during the third trimester, as specified in the trial protocol [22]. The inclusion criteria were individuals who had participated in the IMPACT BCN trial and had no contraindications to MR, such as claustrophobia or metallic implants or devices. A total number of 180 participants accepted and provided written informed consent for maternal brain MR, and we obtained complete images from 123 participants. For the objective of this study, we included the images from the Mediterranean diet group (*n* = 38) and the usual care group (*n* = 43). After excluding the datasets which had a suboptimal reconstruction quality, 34 datasets from the Mediterranean diet group and 37 datasets from the usual care group were analyzed. Figure 1 displays a flowchart of the study population.

In the Mediterranean diet intervention, we aimed to change the overall dietary pattern instead of modifying the intake of specific key foods or nutrients. The intervention was conducted by trained dietitians and consisted of monthly face-to-face individual interviews, a telephone interview every 15 days and monthly group educational sessions until the end of the intervention (34–36 weeks of gestation). The means (SD) of the baseline and the final visits were 22.9 (1.3) weeks and 35.1 (1.6) weeks of gestation. Also, the participants were provided with free extra virgin olive oil (EVOO, 2 L each month) and walnuts (450 g each month). Additionally, specific materials such as recipes, a 1-week shopping list of seasonal food items and a weekly meal plan with detailed menus were provided at each visit.

The usual care group received pregnancy care as per the institutional protocols. Additional details on the interventions are provided elsewhere [22,27].

### 2.2. Maternal Brain MR Acquisition and Processing

Data were acquired using two MR scanners from different manufacturers between 32 weeks 0 days and 39 weeks 6 days of gestation. One was a Siemens (MAGNETOM Trio Tim, Siemens Healthcare, Erlangen, Germany) 3 T system equipped with a 32-channel head coil, and the other was a Philips (Achieva, Philips Healthcare, Best, the Netherlands) 3 T system equipped with a 32-channel head coil. Anatomical images were acquired using high-resolution T1-weighted (T1-w) axial scans using a fast acquisition gradient echo sequence with magnetization preparation (MPRAGE), with the following protocol for Siemens—(repetition time (TR) = 2300 ms; echo time (TE) = 2.08 ms; flip angle (A) = 8°; matrix size = 240 × 240 × 240; voxel side = 0.8 mm^3^)—and a turbo field echo (TFE) sequence with the following protocol for Philips—(TR = 8.1 ms; TE = 3.7; FA = 8°; matrix size = 240 × 240 × 180; voxel size = 1 mm^3^)—used for the data collection. The images were checked by a certified radiologist and discarded if there were quality problems or structural anomalies. The MR was performed at a mean (SD) of 36.4 (0.9) weeks of gestation, with a similar gestational age in both study groups (Mediterranean diet: 36.4 weeks (1.1) vs. usual care: 36.5 weeks (0.8), *p* = 0.54). Around 50% of the participants were examined using the Philips MR scanner in both intervention groups (Mediterranean diet: 18 (52.9%) vs. usual care: 17 (45.9%), *p* = 0.54).

Cortical surface reconstruction was systematically executed using T1-weighted MRI scans from each participant. This process was facilitated by the FreeSurfer software package, specifically using the recon-all pipeline (version 7.1; Athinoula A. Martinos Center for Biomedical Imaging, Charlestown, MA, USA). The technical procedures encompassed several stages: motion correction [28], removal of the skull and extraneous non-brain tissue [29], transformation into Talairach space [30,31] and segmentation of the white matter along with the deep gray matter structures [31,32]. This was followed by intensity normalization [33] and tessellation at the gray matter/white matter boundary, with subsequent automatic topological corrections [30]. Once the reconstructions were generated, any that were deemed unsatisfactory were excluded. Metrics were gauged in alignment with FreeSurfer’s established criteria. Cortical volume was ascertained by subtracting the volume within the white surface from that inside the pial surface, excluding the subcortical components. Cortical thickness was computed as the mean distance between points on the white surface and their closest counterparts on the pial surface [34]. Each vertex’s determination was based on the average area of the surrounding triangles, corresponding to the white surface’s area. The Desikan–Killiany atlas facilitated the extraction of distinct cortical regions of interest (ROIs) for each metric. Beyond these metrics, FreeSurfer also provides an estimate of the total intracranial volume by employing registration-based techniques. This is achieved by linearly transforming each participant’s data into a template, a method detailed by Buckner et al. [35].

### 2.3. Dietary Questionnaires and Biomarkers

All the participants from the IMPACT BCN trial were assessed on their diet habits by dietitians at a baseline visit (20–24 weeks of gestation) and at a final visit (34–36 weeks of gestation) using a 151-item semi-quantitative food frequency questionnaire (FFQ), validated in the present study population [36], and a 17-item dietary screener based on a previously validated Mediterranean diet adherence score adapted to pregnancy [15,27]. The participants indicated their usual and frequency consumption of the listed food items in the FFQ based on nine frequency categories (ranging from never or <1 time/month to ≥6 times/day) and using common units or portion sizes. The 17-item dietary screener was used to assess the Mediterranean diet adherence level, including recommendations for each food’s intake, such as EVOO (≥4 tablespoons/day); vegetables and dairy products (≥3 servings/day); fresh fruit (≥2 servings/day); wholegrain cereals (≥5 servings/week) and legumes, walnuts, fish and white meat (≥3 servings/week). They were advised to limit their intake of red meat and processed meat; carbonated and/or sugar-sweetened beverages; butter, margarine or cream (<1 serving/week) and pastries (<2 servings/week). If a criterion was met, 1 point was given, with the total points ranging from 1 to 17. Improvements in the Mediterranean diet were defined as an improvement of at least 3 points in the final score of the 17-item dietary screener compared with the baseline score.

The concentrations of the selected food biomarkers of EVOO and walnuts were evaluated in a subsample of randomly selected participants (30% from each group; Mediterranean diet group *n* = 19, usual care *n* = 21). Specifically, their urinary hydroxytyrosol levels (for EVOO consumption) and plasma oleic acid, α-linoleic acid and α-linolenic levels (for walnut consumption) were measured at the baseline and at the final visits. Detailed information on each biological sample measurement method is available in the trial protocol [22].

### 2.4. Statistical Analysis

The normal distributions of the variables were tested using the Shapiro–Wilk test and histograms. Student’s *t*-test or the Mann–Whitney test as appropriate for continuous parameters and the chi-square test or Fisher’s exact test as appropriate for categorical parameters were used to assess the differences between the Mediterranean diet and usual care groups in terms of their baseline characteristics.

Brain MR structural evaluations, encompassing volume, thickness and cortical area reconstructions, were standardized to a shared spherical atlas space, facilitating a detailed vertex-by-vertex cluster analysis. Datasets from both the Mediterranean diet and usual care group were juxtaposed using a general linear model, examining metric differences with covariates such as total intracranial volume, age, MR scanner and nulliparity. A secondary model (model 2) incorporated additional variables: initial Mediterranean diet score assessment and socio-economic status. The general linear model outcomes underwent correction for multiple comparisons via the “mri_glmfit_sim” tool, setting a vertex-wise threshold at 1.3 and a cluster-wise *p*-threshold of <0.05 to manage the false discovery rate.

A linear regression model with adjustment model 2 was used to discern the association between significant regions of interest and individual items from the 17-item dietary screener, as well as the levels of biomarkers of olive oil and walnut consumption. Analysis of covariance (ANCOVA) was used to assess the changes in dietary intake obtained from the FFQ, including key food consumption, energy and nutrient intake and fatty acid profile, as well as the food biomarkers at the end of the intervention in each group, by adjusting for the baseline values.

A *p* value of <0.05 was deemed indicative of statistical significance. Statistical comparisons and adjusted means were computed with the emmeans library (v. 1.8.2). The statistical analyses were performed using RStudio (version 1.4.1106, RStudio) with software R (version 4.0.5, R Foundation) and Stata (version 16).

## 3. Results

### 3.1. Characteristics of the Study Population

No differences were found in the maternal baseline or perinatal characteristics of these participants (Table 1). As expected, the FFQ results showed that the participants allocated into the Mediterranean diet group had a higher consumption of key foods related to a Mediterranean diet, including EVOO, walnuts and fish, and a higher intake of PUFAs, including α-linoleic acid, α-linolenic acid, eicosapentaenoic acid and DHA (Appendix A). While the Mediterranean diet score at the initial assessment did not differ between the groups (Table 1 and Appendix A), both the Mediterranean diet score and the rate of improvements in adherence to the Mediterranean diet at the final visit were significantly higher in the Mediterranean diet group (*n* = 24, 7%) (Appendix A).

### 3.2. Maternal Brain MR Results

The mothers from the Mediterranean diet group had significantly larger left superior parietal and right precuneus areas (Figure 2 and Appendix A) as compared to the participants from the usual care group. These differences remained statistically significant even after adjusting for additional variables (model 2) (Appendix A). No differences were found in the cortical thickness or the cortical volume of these areas nor any other brain areas/volumes.

### 3.3. Association of Maternal Brain MR and Diet

Participants with high improvements in the Mediterranean diet showed a significantly larger right precuneus area compared to the participants with low improvements (Table 2). An improvement in the Mediterranean diet was not reflected in statistically significant differences in the superior parietal area.

Specifically, the right precuneus and left superior parietal areas were larger in the participants who fulfilled the criteria of optimal walnut consumption (≥3 servings/week), and a tendency was seen for blue fish consumption (≥3 servings/week) without statistical significance (Table 3 and Appendix A). In addition, both the right precuneus and left superior parietal areas showed positive associations with urinary hydroxytyrosol levels (Table 3).

## 4. Discussion

In this secondary analysis of the IMPACT BCN randomized clinical trial, mothers from the Mediterranean diet group showed larger surface areas in the left superior parietal and right precuneus regions compared to the mothers from the usual care group. The right precuneus area was associated with high improvements in the Mediterranean diet adherence score, and both areas were associated with walnut consumption (≥3 servings/week) and with the biomarker of EVOO intake (urinary hydroxytyrosol). To our knowledge, this is the first study to assess the effect of the Mediterranean diet on the brain morphometry of a pregnant population.

The superior parietal lobe is involved in several brain functions, such as visuomotor, cognitive, sensory, spatial cognition and attention [37,38]. The precuneus is one of the last regions to myelinate, with the most complex columnar cortical organization [39]. It is involved in many complex functions, such as autobiographical memory and cognitive functioning, but a unique fact is that it is involved in assigning a first-person perspective (the viewpoint of the observing self).

Two observational studies in an elderly cohort reported that high consumption of fish was positively associated with the gray matter volume in the precuneus and superior parietal cortex, along with other brain regions [3,7]. Furthermore, a randomized trial with middle-aged participants with a 4-week Mediterranean diet intervention reported that the participants in the Mediterranean diet intervention group increased in their cerebral perfusion in several areas, including the precuneus, compared to the Western diet group [40]. Unfortunately, none of these studies investigated the associations between the morphometrical findings and some key foods in the Mediterranean diet, such as EVOO intake or walnuts.

In this study, we did not find differences in the total gray matter, the frontal cortex or the posterior cingulate cortex, as reported in previous studies related to the Mediterranean diet [3,4,41,42]. This was probably due to the relatively short duration of the intervention (mean of 12 weeks) and moreover the population of pregnant individuals. Hoekzema et al. reported that pregnancy reduces the cortical volumes of the frontal gyrus and the posterior cingulate gyrus, along with the precuneus and many other areas [8]. Among the areas reported to decrease in volume, a larger right precuneus area was found in our Mediterranean diet group compared to the controls. These areas were associated with two key foods rich in (poly)phenols: walnuts, also rich in omega-3 fatty acids and melatonin, and EVOO, also rich in monosaturated fatty acids. Both foods and their corresponding nutrients are associated with several health benefits, including a reduced risk of mortality [43,44]. In the case of walnuts, Ni et al. found improvements in general cognitive function with frequent nut consumption after 2 years of follow-up in older adults at high risk of metabolic syndrome [45]. Aligned with our findings, consuming ≥ 3 servings per week of walnuts showed significant effects. In addition, Sala-Vila et al. reported the potential of walnuts to delay cognitive decline in a high-risk population [46]. Regarding EVOO, a clinical trial in an elderly population at high vascular risk disclosed better cognitive performance in the participants randomly assigned to a Mediterranean diet supplemented with EVOO than in those assigned a control low-fat diet [2].

One hypothesis that might explain the larger precuneus area might be the rich intake of PUFAs in the Mediterranean diet group. Regarding PUFA intake during pregnancy, fish, particularly low-mercury options, should be promoted since they can be an important source of PUFAs [47]. The accelerated neurological development of the fetus requires a high amount of PUFAs, which is a fundamental component of the brain not only of the fetus but also of the mother. The abundant omega-3 PUFA intake in the Mediterranean diet may have resulted in a reduced decline in these areas through its positive effects on phospholipid accumulation and membrane composition [12].

Another potential mechanism may be the neuroprotective effect of the Mediterranean diet due to its anti-inflammatory properties. The Mediterranean diet is characterized not only by a healthy fatty acid composition but also a high intake of antioxidant compounds like vitamins from vegetables/fruits and (poly)phenols from plant-based foods, specifically from EVOO and walnuts [48]. Although a specific mechanism of action cannot be identified yet, the cellular mechanisms underlying the volumetric reduction in the cortex during pregnancy is suspected to be a result of synaptic pruning and myelination, driven by sex hormonal fluctuation [10] in order to adapt to this life-changing event. Synaptic pruning is considered the result of a diverse array of immune signaling mechanisms [49], where the anti-inflammatory effect of the Mediterranean diet may modify the balance of synaptic pruning and myelination.

Significant associations were observed between the urinary levels of hydroxytyrosol and a larger surface area in the left superior parietal and right precuneus regions. The potential protective role of hydroxytyrosol in terms of neurological outcomes has previously been described, mainly by mediating several pathways associated with neurological diseases, including chronic inflammation, oxidative stress, mitochondrial dysfunction, energy metabolism and autophagy [50]. In pregnancy, Yeste et al. observed that hydroxytyrosol supplementation during pregnancy in piglet models with high risk of intrauterine growth restriction influenced catecholaminergic and serotoninergic neurotransmission in several brain areas in the mothers [51].

The major strength of this study is the well-structured intervention in a randomized clinical trial of a pregnant population. Additionally, the participants were assessed with validated questionnaires applied by nutritionists and objective measurements of food intake biomarkers.

The study has several limitations. First, two different MR scanners were used. For this reason, MR scanner was included as a covariate in the adjusted model. Second, we lost around 37% of the datasets due to the mothers’ movement. Although the participants could choose their most comfortable position, their advanced gestational age may have caused the participants difficulty in holding still in the scanners. Third, MR was performed only at the end of pregnancy. Since it was not a longitudinal study, we could not assess the changes during pregnancy. Fourth, the study did not include mother-to-infant attachment. Further research including a mother-to-infant attachment test might reveal the functional meaning of the brain differences found in this study. Fifth, the population of this study does not represent a general pregnancy population due to the inclusion criteria of being at high risk of SGA births. In addition, the study was conducted in a high-resource setting in a population with a low proportion of obesity and gestational diabetes [22]. For this reason, the results of this study may not be replicable in other settings. Last but not least, although the maternal brain morphometry observations were prespecified in the study protocol, the main randomized clinical trial was not designed for this study. Therefore, the findings of this study shall be taken as preliminary. They require further replication in more diverse populations.

## 5. Conclusions

A structured lifestyle intervention based on a Mediterranean diet during pregnancy has a significant effect on maternal brain structure, specifically associated with a larger area of the left superior parietal and right precuneus cortex regions. Both areas were associated with sufficient walnut consumption and a biomarker related to EVOO consumption. Confirming these results with future longitudinal studies in a population at low risk of SGA births and exploring mother–child bonding may provide clues to reveal the functional meaning of these brain differences.

## Figures and Tables

**Figure 1 nutrients-16-01604-f001:**
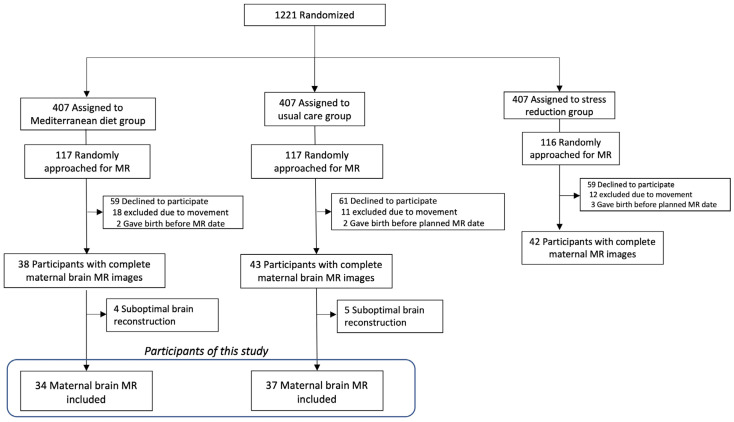
Flowchart of the study population.

**Figure 2 nutrients-16-01604-f002:**
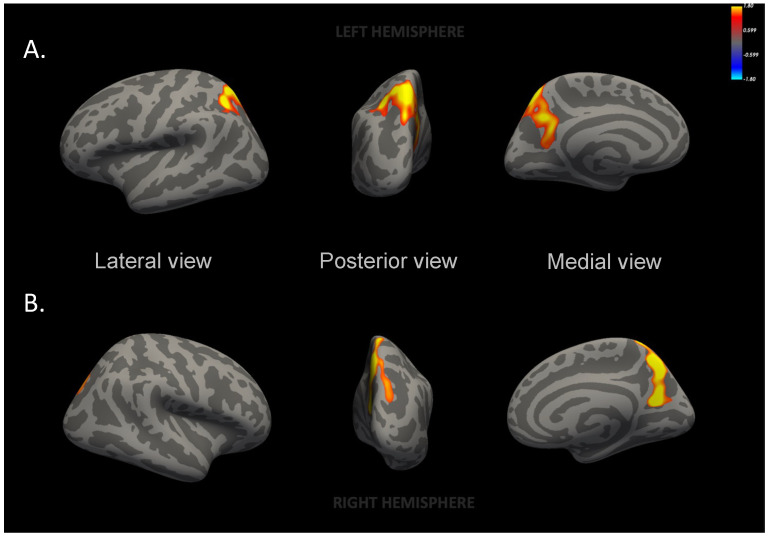
Inflated maps of cortical surface area differences between Mediterranean diet and usual care groups. (**A**) Left superior parietal area, *p* = 0.03; (**B**) right precuneus area, *p* < 0.001. Maternal brain surface in lateral, posterior and medial views. Images generated from a general linear model with total intracranial volume, age, magnetic resonance protocol and nulliparity. The color bar indicates logarithmic scale of *p* values (−log10). Red to yellow color reflects the increased surface area in the Mediterranean diet group participants as compared to usual care group participants.

**Table 1 nutrients-16-01604-t001:** Maternal baseline and perinatal characteristics of participants according to the intervention groups.

Characteristics	Mediterranean Diet	Usual Care	*p* Value
	*n* = 34	*n* = 37	
Age (years)	38.3 (3.3)	37.0 (4.7)	0.17
Race and ethnicity			0.33
African American	1 (2.9%)	0 (0.0%)	
Asian	1 (2.9%)	1 (2.7%)	
Latin	4 (11.8%)	1 (2.7%)	
White	28 (82.4%)	35 (94.6%)	
Education level			0.95
No/primary	1 (2.9%)	1 (2.7%)	
Secondary/university	33 (97.1%)	36 (97.3%)	
Socio-economic status ^I^			0.29
Low	1 (2.9%)	1 (2.7%)	
Medium	12 (35.3%)	7 (18.9%)	
High	21 (61.8%)	29 (78.4%)	
BMI before pregnancy (kg/m^2^)	24.0 (5.0)	22.8 (2.8)	0.19
Previous medical condition			
Thyroid disorders	5 (14.7%)	1 (2.7%)	0.07
Autoimmune diseases	2 (5.9%)	8 (21.6%)	0.06
Diabetes mellitus	6 (17.6%)	2 (5.4%)	0.10
Chronic hypertension	4 (11.8%)	1 (2.7%)	0.14
Psychiatric disorders	1 (2.9%)	3 (8.1%)	0.35
Nulliparous	18 (52.9%)	22 (59.5%)	0.58
Assisted reproductive technologies	13 (38.2%)	9 (24.3%)	0.21
During pregnancy			
Cigarette smoking	5 (14.7%)	5 (13.5%)	0.89
Alcohol intake	6 (17.6%)	7 (18.9%)	0.89
Drug consumption	2 (5.9%)	0 (0.0%)	0.14
Sports practice	11 (32.4%)	10 (27.0%)	0.85
Gestational age at recruitment (weeks)	20.8 (0.7)	20.7 (0.7)	0.43
Mediterranean diet adherence score	8.1 (2.6)	7.8 (2.5)	0.61
Pregnancy outcomes			
Gestational diabetes	4 (11.8%)	2 (5.4%)	0.34
Gestational hypertension	1 (2.9%)	1 (2.7%)	0.95
Preeclampsia	2 (5.9%)	3 (8.1%)	0.71
Preterm birth	0 (0.0%)	0 (0.0%)	0.72

Data are expressed as means (SD) or *n* (%). BMI: body mass index. ^I^ Socio-economic status defined as low if participants reported having never worked or being unemployed for more than 2 years and having a partner with unqualified work or who was unemployed; high if they reported university studies regardless of whether they were working and medium for any other situation.

**Table 2 nutrients-16-01604-t002:** Association between maternal brain areas and improvements in the Mediterranean diet adherence scores at the final visit.

Area	Improvements in the Mediterranean Diet		Adjusted Mean Difference (95% CI)	*p* Value
	High	Low		
	*n* = 30	*n* = 41		
Left superior parietal	5221 (95.6)	4992 (87.3)	228.9 (−25.0–482.8)	0.07
Right precuneus	3691 (62.2)	3513 (56.8)	177.2 (12.0–342.4)	0.03

Area data are expressed as estimated marginal means (SE) (mm^2^). CI: confidence interval. Values were generated from a regression model adjusted for nulliparity, age, scanner, socioeconomic status and the Mediterranean diet adherence score at the final evaluation. The improvements in the Mediterranean diet adherence score were obtained at the final visit, at a mean (SD) of 35.1 (1.6) gestational weeks.

**Table 3 nutrients-16-01604-t003:** Association between maternal brain areas, the dietary 17-item questionnaire and biomarkers obtained at the final visit.

	Left Superior Parietal Area		Right Precuneus Area	
	Adjusted Mean Difference (95% CI)	*p* Value	Adjusted Mean Difference (95% CI)	*p* Value
**Dietary 17-Item Questionnaire ^I^**				
Extra virgin olive oil	191.2 (−190.7–573.1)	0.32	208.4 (−39.0–455.8)	0.09
Vegetables	−12.0 (−273.4–249.4)	0.93	−46.3 (−217.7–125.0)	0.59
Fruits	104.3 (−178.8–387.5)	0.46	99.7 (−85.4–284.9)	0.28
Sofrito	−33.3 (−284.5–217.8)	0.79	−25.8 (−190.8–139.2)	0.75
Wholegrain cereals, bread, pasta	223.1 (−31.0–477.2)	0.08	−5.1 (−176.0–165.9)	0.95
Refined cereals, bread, pasta	207.4 (−57.4–472.1)	0.12	125.4 (−49.1–299.8)	0.15
Legumes	108.9 (−153.7–371.4)	0.41	48.3 (−124.7–221.3)	0.57
Fish/seafood	−68.5 (−317.9–180.9)	0.58	−85.3 (−248.1–77.5)	0.29
Fatty fish	286.0 (−4.3–576.3)	0.05	115.6 (−78.8–309.9)	0.23
Red meat	−16.5 (−333.4–300.4)	0.92	−34.7 (−242.8–173.3)	0.73
Processed meat	121.8 (−177.4–421.1)	0.42	17.2 (−180.4–214.8)	0.86
Chicken, turkey, rabbit, lean pork	−33.7 (−293.6–226.1)	0.80	−93.8 (−263.0–75.3)	0.27
Carbonated and/or sugar-sweetened beverages	38.7 (−234.5–311.9)	0.78	38.6 (−140.7–218.0)	0.66
Nuts, including walnuts, almonds, peanuts	375.7 (78.9–672.4)	0.01	210.8 (13.2–408.4)	0.03
Pastries such as cookies, custard pastries or cake	−7.0 (−266.8–252.9)	0.96	1.3 (−169.4–172.1)	0.98
Dairy products, including calcium-fortified vegetable milk	154.1 (−100.1–408.3)	0.23	−45.3 (−213.9–123.2)	0.59
Butter, margarine or cream	73.7 (−188.9–336.3)	0.58	28.0 (−144.8–200.7)	0.74
**Biomarkers ^II^** **(μmol/g creatinine)**	β (95%CI)	*p* value	β (95%CI)	*p* value
Oleic acid	−1.2 (−52.3–49.9)	0.96	−19.7 (−52.5–13.0)	0.24
Alpha-linolenic acid	−113.6 (−1161.6–934.5)	0.83	−83.0 (−768.3–602.3)	0.81
Alpha-linoleic acid	−1.3 (−39.4–36.9)	0.94	3.8 (−21.2–28.7)	0.77
Hydroxytyrosol	119.3 (39–199.6)	0.006	74.3 (21.3–127.4)	0.01

^I^ Contrast for dietary 17-item questionnaire was assessed as 1 point–0 point. ^II^ Number of participants available for biomarkers: Mediterranean diet group: *n* = 19, usual care: *n* = 21. Values were generated from a regression model adjusted for nulliparity, age, scanner, socio-economic status and the Mediterranean diet score at the initial assessment. The dietary information was obtained at the final visit, at a mean (SD) of 35.1 (1.6) gestational weeks.

## Data Availability

The data described in the manuscript, code book and analytic code will be made available upon request, with approval by the ethical committee of the author’s institute and a signed data access agreement. To access these data, one should contact francesca.crovetto@sjd.es via email.

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
