# Peer review of "The Mediterranean Diet in Pregnancy: Implications for Maternal Brain Morphometry in a Secondary Analysis of the IMPACT BCN Randomized Clinical Trial"

_nutrients, 2024, doi:10.3390/nu16111604_

Round 1
Reviewer 1 Report
Comments and Suggestions for Authors
nutrients-2990047
The article “Mediterranean diet in pregnancy: implications for maternal 2 brain morphometry. A secondary analysis of the IMPACT BCN 3 randomized clinical trial” reports the results of maternal brain MRI in women during pregnancy who were participating in the IMPACT BCN RCT. The study is interesting; however major revisions are suggested prior to consideration for publication to improve the clarity and understanding of the study. Additionally, careful English language editing is suggested.
Introduction
1. Lines 64-65: Please revise for clarity – I think a word or part of the sentence is missing.
2. Lines 82-85: Please justify the recommendation of 300 mg/day of PUFA during pregnancy. Existing guidelines recommend 200 mg/day of omega-3, but not of omega-6. The paper cited provides information on serum concentrations of fatty acids, not on intake requirements. Please specify if this is only for omega-3.
Methods
3. Overall: Please ensure that there is clarity and transparency throughout the methods. Please also ensure that numbers of participants are reported clearly as each stage of the study. It seems that the main results center on the n=34 and n=37 participants that had brain imaging done; however, it seems that fewer participants had biomarker samples taken.
4. Lines 180-181: It seems that the Food Frequency Questionnaire was not used at all for this study. Suggest removing it from the methods section, as it creates confusion.
5. Line 181: Please provide the full citation for the 17-item dietary screener. Additionally, please provide brief information on how the screener was adapted for pregnancy and what screening tool it was based off of.
6. Lines 191-193: The term “high adherence” in this case seems misleading, as the authors are referring to an improvement in adherence to the Mediterranean diet. If a participant had a very low baseline Mediterranean diet score, an increase of 3 points would not necessarily confer the designation of high adherence. Suggest the authors revise this to “improvements in Mediterranean diet” or some other similar designation.
7. Lines 217-220: It seems the results pertaining to this analysis are missing. Please remove this section or add results.
Results:
8. Lines 228-240: This information would be better suited for the methods section, as it provide the contextual information of numbers of participants throughout the study, which helps the reader interpret the results. Information describing the results of table 1 generally go under characteristics of the population.
9. Table 1: Please add the mean (SD) Mediterranean diet score at baseline. Also, please revise “study class” to “education level”
10. Lines 272-274: Please clarify that the changes in left superior parietal area were not statistically significant.
11. Lines 282: By your criteria, <0.05 is considered statistically significant, but results related to fish have a p value=0.05. If this changes how the results are reported, please ensure the discussion reflects the revised results.
12. Table 2 and Table 3: Please clarify in the titles whether dietary information is baseline or follow-up (and mean gestational age at assessment of diet).
Discussion:
13: Overall: The authors reported that the Mediterranean diet scores were not significantly different at baseline between mothers in the Med diet group and the usual care group. Can we truly say that the differences seen in brain images are the result of the short-term improvements in adherence to a Mediterranean diet? Could there be other alternate explanations?
14: Lines 307-310: This section is difficult to understand. Please revise for clarity.
15: Lines 313-331: This section seems to suggest that the results might be explained by polyphenol intake rather than PUFAs. Suggest making this clearer as an alternate explanation for the results.
16: Lines 332-333: The authors claim that the rich PUFA intake in the Mediterranean diet group may explain the results. Suggest providing information from the FFQs to show this. Your biomarker results showed that alpha linolenic acid was not associated with brain images. This, of course, does not include potential of higher DHA or EPA levels from fish intake, but, this is hard to justify without data or information on the kinds of fish participants were consuming.
17. Guidelines for fish consumption during pregnancy encourage women to consume no more than 1-2 servings/week to avoid excess mercury intake. Some of the higher omega-3 fish options are also higher in mercury and therefore, may have been avoided. Please comment on this in the discussion.
Comments on the Quality of English Language
This paper will benefit from careful English language revisions. At times it is difficult to understand.
Reviewer 2 Report
Comments and Suggestions for Authors
This is a well-written and well-organized research article with adequate novelty and quality. Some points should be addressed.
Abstract
- The authors should add subheadings according to the journal guidelines.
- The abstract is a bit long. The authors should try to condense the methods description. The should also try to keep the most significant results by omitting some of them.
Introduction
- The second paragraph is quite long. The authos could split ths paragraphs into two smaller paragraphs.
- The authos should more emphasize the literature gap that aim to cover with the present study.
Methods
- This part of manuscript is very well-written and very well-organized.
Results
- The resolution of Figure 1 should be improved and enlarged.
- The words and the numbers in tables should be decreased according to the journal guidelines.
Discussion
- This part of manuscript is very well-written and very well-organized, including a useful description of the strengths and the limitations of the study and performing a well comparison analysis with previous relevant studies on the field.
Conclusions
- The authors should emphasize their opinion based on their expertise and the current results of their study concerning what future studis could be useful to be performed to cover further the topic of their study.
Comments on the Quality of English LanguageMinor editing of English language is recommended.
Round 2
Reviewer 1 Report
Comments and Suggestions for Authors
The authors have adequately addressed my comments and the manuscript has been significantly improved.